# Protocol on a multicentre statistical and economic modelling study of risk-based stratified and personalised screening for diabetes and its complications in India (SMART India)

Sobha Sivaprasad [1,2] Rajiv Raman [3] Ramachandran Rajalakshmi,[4] Viswanathan Mohan,[4] Mohan Deepa,[4] Taraprasad Das,[5] Kim Ramasamy,[6] A Toby Prevost [7] Raphael Wittenberg,[8] Gopalakrishnan Netuveli,[9] Gopal Lingam,[10] Wasim Hanif,[11] Radha Ramakrishnan,[2] Jayashree Ramu,[1] Janani Surya,[3] Dolores Conroy,[2] On behalf of SMART India Study Collaborators

**Correspondence to**
Dr Sobha Sivaprasad;
s.sivaprasad@ucl.ac.uk

## ABSTRACT

**Introduction** The aim of this study is to develop practical and affordable models to (a) diagnose people with diabetes and prediabetes and (b) identify those at risk of diabetes complications so that these models can be applied to the population in low-income and middle-income countries (LMIC) where laboratory tests are unaffordable.

**Methods and analysis** This statistical and economic modelling study will be done on at least 48 000 prospectively recruited participants aged 40 years or above through community screening across 20 predefined regions in India. Each participant will be tested for capillary random blood glucose (RBG) and complete a detailed health-related questionnaire. People with known diabetes and all participants with predefined levels of RBG will undergo further tests, including point-of-care (POC) glycated haemoglobin (HbA1c), POC lipid profile and POC urine test for microalbuminuria, retinal photography using non-mydriatic hand-held retinal camera, visual acuity assessment in both eyes and complete quality of life questionnaires. The primary aim of the study is to develop a model and assess its diagnostic performance to predict HbA1c diagnosed diabetes from simple tests that can be applied in resource-limited settings; secondary outcomes include RBG cut-off for definition of prediabetes, diagnostic accuracy of cost-effective risk stratification models for diabetic retinopathy and models for identifying those at risk of complications of diabetes. Diagnostic accuracy inter-tests agreement, statistical and economic modelling will be performed, accounting for clustering effects.

**Ethics and dissemination** The Indian Council of Medical Research/Health Ministry Screening Committee (HMSC/2018–0494 dated 17 December 2018 and institutional ethics committees of all the participating institutions approved the study. Results will be published in peer-reviewed journals and will be presented at national and international conferences.

**Trial registration number** ISRCTN57962668 V1.0 24/09/2018.

---

### Strengths and limitations of the study

► This is the first national prospective study that will assess the prevalence of sight-threatening diabetic retinopathy (DR) in various regions in India.
► The study will provide evidence on the accuracy of point-of-care glycated haemoglobin as a screening tool for diabetes.
► The study will provide several diagnostic models on diabetes and its complications.
► Validation of the models may not be possible in all cases.
► The treatment pathway for patients identified with sight-threatening DR or other complications of diabetes is according to local protocols.

## INTRODUCTION
### Background
Diabetes and its complications are common causes of morbidity and mortality globally. Low-income and middle-income countries (LMIC) are most affected by the diabetes epidemic, where significant number of people with undiagnosed diabetes present with complications of diabetes.[1] More than 30% of world population is estimated to have prediabetes.[2] The most common risk factors for diabetes and its complications are long-term diabetes, uncontrolled hyperglycaemia, hypertension and dyslipidaemia. As high as 90% of people with type 2 diabetes are dyslipidaemic and 60%–85% are hypertensive. In addition, 90% of people with type 2 diabetes are obese.[3] There is an unmet need to screen for prediabetes and diabetes in LMIC, where primary healthcare is underdeveloped and laboratory tests are costly.

## Screening for people at risk of diabetes

According to the WHO, diabetes is confirmed by laboratory tests in a symptomatic individual if glycated haemoglobin (HbA1c) is ≥48 mmol/L (≥6.5%) or fasting blood glucose is ≥7 mmol/L (≥126 mg/dL), or a random blood glucose (RBG) is ≥11.1 mmol/L (≥200 mg/dL) or after a 2-hour oral glucose tolerance test, blood glucose is ≥11.1 mmol/L (≥200 mg/dL). In asymptomatic individuals, diabetes has to be confirmed by two of these laboratory tests.[4] Standard laboratory-based HbA1c test has the added advantage of providing an average estimation of the glycaemic status of an individual over the previous 3 months and is helpful in categorising people into normal (HbA1c <42 mmol/mol; <6.0%), prediabetes (HbA1c 42–47 mmol/mol; 6%–6.4%) and diabetes (HbA1c is ≥48 mmol/mol; ≥6.5%).[4] The lower limit of HbA1c in prediabetes may be as low as 5.7%.[5]

However, none of these tests are practical for population-level screening in LMIC where non-technical personnel often conduct screening for diabetes in non-clinical environments. HbA1c also cannot be measured in patients with haemoglobinopathies. A number of LMIC have high prevalence of malaria and various haemoglobinopathies, including thalassaemia and sickle cell anaemia. Therefore, there is an unmet need to use simple tests to identify people at risk for diabetes. Despite its variability, capillary RBG is the the most common blood test done in such situations.[6] Prediabetes is not clearly defined by RBG despite several studies that have attempted to define cut-off values of RBG against HbA1c.[6–15] More convenient point-of-care (POC) HbA1c kits are now available that show good correlation with laboratory-based HbA1c estimation.[16] It is, therefore, appropriate to validate POC HbA1c against RBG in community screening. Although there are several studies that have evaluated various screening tests for prediabetes, these studies have used laboratory-based HbA1c measurements or fasting blood glucose as the index test.[17] In contrast, this study will focus on POC HbA1c as the index test for prediabetes to inform community screening. Studies using POC HbA1c as a reference test have included specific disease cohorts only, or had a small sample size within hospital settings or conducted post-hoc analysis on previously recruited study cohorts and most importantly, did not compare the accuracy of these tests with known non-laboratory (NL) based diabetes risk scores.[6–15]

Due to the large numbers of undiagnosed diabetes, it is also useful to investigate whether it is more efficient to triage people at risk of diabetes in the population using non-invasive diabetes risk scores, such as Madras Diabetes Research Foundation-Indian Diabetes Risk Score (MDRF-IDRS)[18] to further reduce the cost of screening with POC HbA1c or RBG.

## Screening for complications of diabetes mellitus

Approximately 30% of people with diabetes present with macrovascular complications such as cardiovascular, cerebrovascular and peripheral vascular diseases.[3] In addition, this population may also have microvascular complications, including diabetic kidney disease (DKD) in 30%–50%, diabetic retinopathy (DR) in 30% and diabetic neuropathy in 30%–50%.[3] Despite this public health burden, people with diabetes are not systematically screened for these complications of diabetes in LMIC due to economic constraints, paucity of public health programmes, inadequately trained manpower and under-resourced infrastructure. Recently, several cardiovascular risk scores such as the NL INTERHEART risk score (IHRS) have been successfully used in community screening programmes.[19] It may be possible to develop similar models to identify people at risk of sight-threatening DR (STDR) and blindness. Although systematic annual photographic retinal screening after pupil dilatation using standard costly retinal cameras and prompt treatment of STDR have reduced the rate of blindness in the UK,[20] these complex and costly screening protocols are not translatable to LMIC and hence alternative screening methods must be considered to ensure population coverage. There are recent reports of accuracy of identifying STDR from the retinal images obtained by affordable and portable non-mydriatic cameras and graded either manually or by artificial intelligence.[21 22] Therefore, adding retinopathy screening, using these hand-held retinal cameras, to minimally invasive tests, such as blood pressure (BP) and urine dip test for microalbuminuria and other NL risk scores, may be an efficient and cost-effective screening option to identify people at risk of diabetes complications.

## Objectives

Our study has three important objectives. The first objective is to determine the ideal tests that could identify people at risk of diabetes and prediabetes in community screening that can be applied to LMIC. In order to accomplish this, we would evaluate the correlation of RBG levels with POC HbA1c levels and decide on a cut-off value for RBG from HbA1c to diagnose prediabetes. Second, we will evaluate whether initial triaging with NL diabetes risk score followed by either RBG or POC HbA1c only to the identified risk group is more effective than screening everyone for diabetes using either RBG or POC HbA1c. Third, we will develop affordable, easily deliverable and clinically effective model to accurately identify people at risk of complications of diabetes in community screening, especially DR.

Secondary objectives are aimed at guiding future policies on screening of diabetes and its complications. As the study involves a large sample and the setting up of a teleophthalmology model to screen for DR across 20 regions in India, we will be able to report the regional prevalence of DR and the associated risk factors, the inter-grader reliability and the accuracy of using artificial intelligence to grade DR. We will also conduct economic modelling and process evaluation of a holistic model for screening of all complications of diabetes. If sample size permits, we will be able to report on region-specific and diverse population-specific rates of diabetes and complications,

visual impairment, quality of life and risk models specific to regions to inform local health authorities.

## METHODS AND ANALYSIS
### Study design

This is a statistical and economic modelling study that will be done on cross-sectional and prospectively recruited participants from community-based screening in order to accurately identify people at risk of diabetes, prediabetes and complications of diabetes.

### Study setting

This community screening will be conducted across 20 regions in India, each led by a local clinical centre with a trained ophthalmologist responsible for the study at that site (figure 1). Each region will have three clusters stratified into urban, rural and a predefined special category of population such as people with poor access to healthcare, or persons that are presumed to be at high risk or low risk of developing diabetes. The study will involve a door-to-door survey, with questionnaires and POC tests performed by fieldworkers. Each cluster will screen at

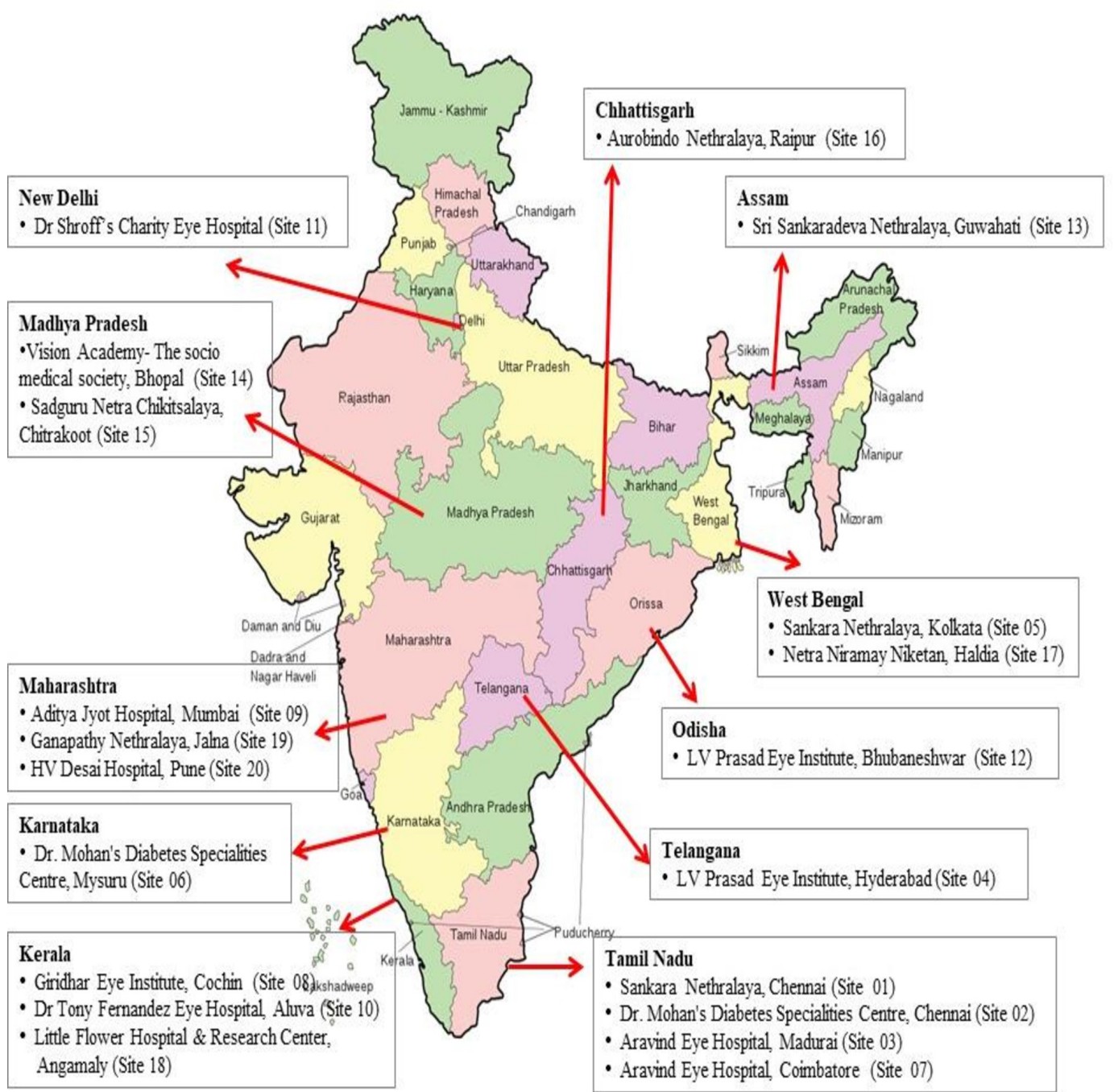

**Figure 1**  Map of India with 20 centres marked.

least 800 consenting individuals aged 40 years or above for a cumulative sample size of a minimum of 48 000 participants. If any cluster or centre does not reach their target recruitment, it will be made up by another cluster or centre with the same stratified population.

### Stratified sampling

In each region, we predefined a geographic area as urban or rural based on a multistage sampling technique using data from the 2011 census of India. A census enumeration block that usually consists of 125–150 households with a population of 650–700 is the primary sampling unit for urban areas, while villages are defined in the rural areas. Bigger villages are further divided to ensure that approximately 300 households can be covered. The house-to-house survey will be conducted by approaching each household in consecutive streets in each area. If the household members are not available, a further two visits by the fieldworkers are permitted. In each household, all available members aged 40 years or above, who meet the inclusion criteria, will be invited to participate in the study.

The special category groups include two groups: (a) people working under high stress leading to poor and untimely eating habits (such as policemen, truck and taxi drivers, manual labourers, fishermen, factory staff and professionals in stressful jobs) and those presumed to be at low risks such as certain religious groups and (b) people with poor health seeking behaviour and/or under social stigma (such as tribal, slum population and people with infection like HIV or leprosy). All survey clusters and special groups are independent samples. The total population for the study is the total recruited participants in all the 20 regions, including the special population (figure 1).

### Selection of participants

The inclusion criteria are adults who are ≥40 years of age (special groups may contain adult population of any age), who are local residents of Indian origin and are willing to give informed consent (see online supplemental appendix 1 for sample informed consent form).

Exclusion criteria include vulnerable adults in whom it may not be possible to carry out all the tests; pregnant and breastfeeding women; anyone in the opinion of the fieldworkers deemed too ill to be screened; and those who are currently participating in intervention trials with investigational medicinal products.

### Study procedures

The fieldworkers will be responsible for providing adequate information about the study and obtaining consent from willing participants. A unique patient identification number will be allocated for each participant to ensure anonymity. A detailed case report form containing a structured questionnaire will be answered by all participants in the study (see online supplemental appendix 2 for case report form). The data collected will include age,

gender, marital status, socioeconomic status (education, occupation and average monthly income), MDRF-IDRS and IHRS that contain questions on lifestyle (smoking and alcohol habits, diet and physical activity and stress),[6 7] brief medical and ocular history with any relevant medications and/or surgery, and family history of diabetes and cardiovascular disease. The structured questionnaire will be translated into local languages and administered by trained fieldworkers. Questionnaires will be validated in 200 subjects in 2 study sites at the start of the study and the case report forms and the study database will be refined to ensure generalisability and reproducibility.

Anthropometric measurements will be performed using the same kits supplied to all sites, and local fieldworkers will be trained on regular calibration of the kits. Height (in cm) will be measured using a stadiometer (SECA Model 214, Seca Gmbh Co, Hamburg, Germany). Weight (in kg) will be measured with an electronic weighing scale (SECA Model 807, Seca Gmbh Co, Hamburg, Germany) kept on a firm horizontal flat surface. Body mass index will be auto-calculated. Waist circumference will be measured at the smallest horizontal girth between the costal margins and the iliac crest at the end of expiration using a non-stretchable measuring tape. Hip measurement will be done with the arms relaxed at the sides, at the maximum circumference over the buttocks.

BP will be recorded in sitting position in the right arm to the nearest 1 mm Hg using the electronic OMRON machine (Omron Corporation, Kyoto, Japan). Participants with BP ≥140/90 mm Hg and not on antihypertensive drugs will be advised to contact a physician for further evaluation. A simple finger-prick test will be used to assess capillary RBG using a standard POC testing device (OneTouch Verio Glucometer, LifeScan Inc, USA). All participants with known diabetes or those with capillary RBG ≥160 mg/dL and 50 participants with RBG 110–159 mg/dL in each cluster will receive further tests. These include HbA1c estimation using a POC kit (A1c Now Plus, PTS Diagnostics, USA) and POC lipid estimation (Cardiochek PA analyser, PTS Diagnostics, USA). A POC urine sample (Chemstrip Micral dipstick, Roche Diagnostics, Mannheim) will be tested for the presence or absence of microalbuminuria.

Visual acuity in both eyes will be recorded using a tablet/smartphone-based vision check web-based application (Peek Vision). Non-mydriatic fundus photography of both eyes will be done using a handheld retinal camera (Visuscout 100, Zeiss, Germany). This portable and battery-operated camera with in-built Wi-Fi facilities will allow capture of colour and red-free retinal images covering 40° field of view through pupils as small as 3.5 mm. Two fundus images (one macula-centred and one disc-centred) of each eye will be captured. In case of any media opacities making fundus imaging difficult, the anterior segment image of each eye would be taken. A teleophthalmology system will be set up whereby the images captured by each fieldworker will be uploaded to a cloud-based study specific database and graded at

the local clinical centre by an ophthalmologist/optometrist (primary grader), as well as transferred to four central reading centres, where grading will be done by a second ophthalmologist (secondary grader). Discrepancies between primary and secondary grading will result in arbitration by a senior retinal consultant. Any participants with STDR, ungradable images and other incidental findings requiring further evaluation will be informed by the fieldworkers and counselled to attend hospital eyecare service. DR will be classified as per the International Clinical Disease Severity Scale for DR as no DR, mild/moderate/severe non-proliferative DR (NPDR) and proliferative DR (PDR).[23] Diabetic macular oedema (DMO) will be determined as present or absent. STDR would be defined as the presence of severe NPDR, PDR and/or DMO. Artificial intelligence may be applied to grade these images and if found to be as accurate as human graders, it will be incorporated to the screening model.

The well-established and widely used quality of life questionnaire EQ-5D (Euro Quality of life) will also be administered with additional vision 'bolt-on' questions and vision-related quality of life.[24–26] The study flow is shown in figure 2. In addition, centre administrators at each clinical site will be responsible for contacting, by letter or phone, and tracking follow-up of those participants who need further referral to an eye hospital for treatment for STDR or due to ungradable retinal images.

## Quality assurance

Training of research personnel on study assessments will be done at study initiation meetings, where the core study team, laboratory staff and camera manufacturers will certify individual fieldworkers. In addition, the data manager in the UK will provide on-site training at each centre, as well as continuous remote training throughout the study. The ophthalmologists or their representatives at each clinical centre will be responsible for training their team who may not meet the pre-set criteria or any new member joining the team. A monitoring plan will be in place to ensure that regular remote monitoring is done throughout the study period.

## Quality control

Calibration procedure and frequency for the weighing machine, BP apparatus, POC kits for capillary RBG and HbA1c and urine will be followed at all centres to avoid any bias or errors. All personnel involved in the grading of retinal images must have completed a study-specific training course.

## Data management

The data will be entered directly by the fieldworkers into a tablet that is linked to a cloud-based electronic database hosted in India. In situations where internet access is not available, paper case report forms will be used at the site and later transcribed into the database. The data in the database will be monitored by the study monitoring team.

The retinal photographs will also be uploaded to the platform. The WHO STEPwise approach to surveillance will be used to develop the cloud-based electronic database.[27] The study is monitored by an independent committee and the progress of the study is reviewed by the grant executive committee.

## Database functionality and quality assurance

The study electronic database (Playon, Bangalore, India) will be hosted on a dedicated secure server in India. All data will be managed through this system. The database will be programmed to perform validation checks, such as range checks to prevent data entry errors, missing data to be flagged up to ensure completion of the data entry. The system will provide for data security and also have formal database lock functionality and it will support real time data cleaning and reporting.

## Statistical considerations

The statistical methods will be developed fully within a statistical analysis plan, to be finalised before database lock. Diagnostic accuracy publications will follow recognised Standards for Reporting Diagnostic accuracy studies guidelines and the observational component will follow the Strengthening the Reporting of Observational Studies in Epidemiology guidelines. Table 1 shows the reference and index tests for diagnostic accuracy aspect of the study.

Accuracy will be measured by sensitivity and specificity of tests to detect diabetes, prediabetes and people at risk of complications of diabetes. Clustering will be used to accommodate any over dispersion. Consistency of these statistics will be explored across centres and clusters (urban, rural and special population). Area under receiver operating characteristic (ROC) curve will be used to compare models representing the overall performance of tests under comparison. Refinement of test components (eg, combinations of tests or questionnaire items) will be developed, and internally validated where sufficient data are available. The number of false positives will be identified directly from the data. From the estimates of sensitivity and the specificity of diabetes risk score to detect prediabetic (or diabetic) and its estimated prevalence, it will be possible to estimate the false positive rate and the complement of the positive predictive value. All estimates will be accompanied by estimated 95% CIs, which account for both clustering and stratification.

For the modelling framework, a marginal model with a logit link will be used, with retinal photograph determination of the reference outcome. Model-predicted probabilities will enable the area under the ROC curve to be estimated with 95% CI allowing for clustering, and accompanied by estimates of sensitivity, specificity, predictive values and likelihood ratios. Diabetes alone, and diabetes or prediabetes will be explored, as will already-identified and newly identified diabetes. For research questions on the diabetes diagnostic model, the denominator will principally be all those diagnosed with diabetes, whether

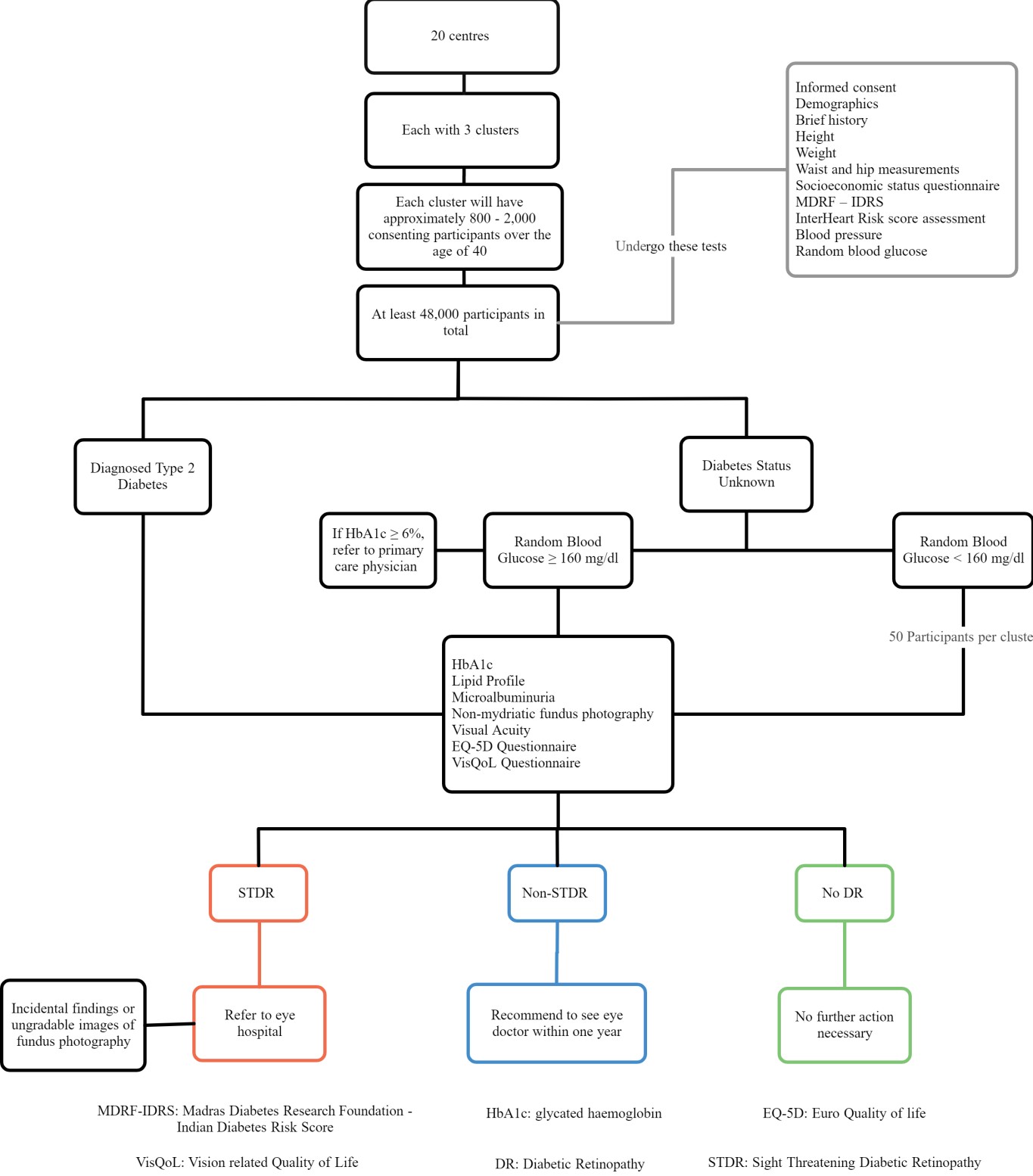

**Figure 2** Study flow diagram.

already diagnosed or newly diagnosed. Interaction with this term (known vs newly diagnosed) will contribute to the analysis involving costs. Further modelling will explore use of the data from those that were found not to have diabetes or prediabetes.

Marginal logistic modelling will be used to identify the tests and questionnaire items which are most predictive, following a recommended approach.[28] Continuous predictors will be handled using the fractional polynomial approach.[29] In the sample size section, it can be seen that the dataset is large enough to allow models to assess up to 10 (reliably) and 20 (less reliably) dependent on intra-cluster correlation. Differences in area under the ROC curve and differences in specificity for given sensitivity

**Table 1** Reference and Index tests

| Community screening for diabetes | |
| --- | --- |
| Reference standard | Index test |
| 1. RBG | 1. POC HbA1c<br>2. Non-invasive diabetes risk scores |
| Community screening for prediabetes | |
| 1. POC HbA1c | 1. RBG<br>2. Non-invasive diabetes risk scores |
| Community screening for complications of diabetes | |
| 1. Serum lipid profile<br>▶ TC<br>▶ Non-HDL cholesterol<br>▶ LDL cholesterol<br>▶ HDL cholesterol<br>▶ TC:HDL ratio<br>▶ Triglyceride<br>2. HbA1c or RBG<br>3. Microalbuminuria<br>4. Retinal photography for retinopathy for all people with diabetes | Risk-based screening tool for complications of diabetes using minimally or non-invasive tests |

HbA1c, glycated haemoglobin; HDL, high density lipoprotein; LDL, low density lipoprotein; POC, point-of-care; RBG, random blood glucose; TC, total cholesterol.

will be estimated. The sample size is large enough to assess existing tests and to develop models. There may be limited scope to validate models. However, interim analysis will allow assumed rates and numbers to be assessed; the number of cases with STDR will be estimated more accurately, and this may enable more sophisticated forms of internal validation. Model validation would include calibration after model discrimination.[30] Clustering within estimates of sensitivity, specificity and areas under the ROC curves will account for clustering, considering use of the non-parametric stratified bootstrap. A similar approach will be undertaken for the model to identify people at risk of complications of diabetes. Models for DR will also test the accuracy of artificial intelligence graded images compared with human graders.

## Sample size calculation
The sample size is determined by considering the numbers of expected STDR, as this analysis will have the smallest number of cases with the outcome. With 20 regions, we expect 216 cases of STDR. From 48 000 people (2400 per centre) screened, of whom about 4800 are expected to be known diabetes and, we suspect, another 4800 will be newly detected diabetes. As 30% of the former group, and 15% of the latter group, are expected to have DR, we anticipated 2160 people to have DR, of whom 216 to have STDR.

Considering that some patients would come from the same family, and some from the same area, we assumed that outcomes at the area level would have an allowed intra-centre correlation (ICC) coefficients of approximately up

to 0.05 and to 0.10 for new and known diabetes, respectively. At the area level, with approximately 100 cases per region, and a working ICC of 0.075, we expect a design effect of 8.5. This calculation has been based on conservative allowances and approximations, which allow for deviations in the actual intracluster correlation coefficients from those anticipated, or for variation in the actual number of cases across centres. This means that the effective sample size (were the sample to be free from clustering) is 25 STDR cases for covariates, which are constant at the region level, or highly correlated among families within the same area. Using the rule of 10 people per covariate in order to plan the number of possible covariates, this implies that it will be possible to include 10–20 covariates (216/10) at the participant-level dependent on whether there is no, modest or moderately high ICC in the covariates, and 1–2 covariates (25/2) either at the area/family level for a stable diagnostic STDR model. All models will include observations at the participant level in order to accommodate participant-level covariates and will accommodate clustering further by including two area contrast terms; these reflect whether a participant lives in the strata of regions that are urban, rural or a special population. Models will be from the 'marginal' class so that correlation can be accommodated while importantly retaining a participant-specific interpretation of resulting estimates. The study will continue to recruit to enable process evaluation and other substudies to be incorporated.

## Health economics analysis plan
The health economics modelling will address the following three questions: (1) What is the cost-effectiveness of a new screening pathway for diabetes and prediabetes? The screening approaches will comprise: diabetes risk score followed by definitive laboratory tests; diagnostic model which the statistical modelling finds to be more accurate than diabetes risk score followed by definitive laboratory tests; RBG for all without diabetes risk score based prescreen; HbA1c test with no prescreen and; no screening; (2) What is the cost effectiveness of a new screening pathway for DR among people with diabetes? The screening approaches will comprise a new method, which the statistical modelling finds to be accurate; retinal photographs only and no screening and (3) What is the cost effectiveness of a new screening pathway for a range of other complications of diabetes among people with diabetes? The screening approaches will comprise a new method, which the statistical modelling finds to be accurate; a combination of HbA1c, lipids and urine tests and colour retinal images; and no screening. In each case, therefore, one comparator will be a 'gold standard' (HbA1c test, retinal photographs and combination of tests as above) and another will be no screening and no treatment until symptoms of DR, DKD or other complications of diabetes are experienced.

The modelling will draw on the following data sources: (1) the data collected through the house-to-house

| Table 2 | SMART India collaborators | | |
|---|---|---|---|
| **Site no.** | **Name of principal investigator** | **Hospital name** | **Ethics approval and date** |
| 1 | Dr Pramod Bhende<br>Dr Rajiv Raman | Sankara Nethralaya, Chennai, Tamil Nadu | Vision Research Foundation, Institutional Review Board<br>Study code: VRF/674A-2018-P<br>Date of approval: 22 March 2018 |
| 2 | Dr Ramachandran Rajalakshmi<br>Dr Viswanathan Mohan | Dr Mohan's Diabetes Specialities Centre, Chennai, Tamil Nadu | Madras Diabetes Research Foundation, Institutional Ethics Committee<br>Date of approval: 6 March 2018<br>Reference number: MDRF/NCT/02–01/2018 |
| 3 | Dr Kim Ramasamy | Aravind Eye Hospital, Madurai, Tamil Nadu | Aravind Medical Research Foundation, Institutional Ethics Committee<br>Reg. number: ECR/182/Inst/TN/2013/RR-19<br>IRB2018010BAS<br>Date of approval: 21 April 2018 |
| 4 | Dr Taraprasad Das<br>Dr Padmaja K Rani | LV Prasad Eye Institute, Hyderabad, Telangana | LV Prasad Eye Institute, Ethics Committee<br>Reference number: LEC07-18-096<br>Date of approval:19th July 2018 |
| 5 | Dr Rupak Roy<br>Dr Supita Das | Sankara Nethralaya, Kolkata | Vision Research Foundation, Institutional Review Board<br>Study code: VRF/674A-2018-P<br>Date of approval: 22 March 2018 |
| 6 | Dr Deepa Mohan | Dr Mohan's Diabetes Specialities Centre, Mysuru, Karnataka | Madras Diabetes Research Foundation, Institutional Ethics Committee<br>Date of approval: 6 March 2018<br>Reference number: MDRF/NCT/02–01/2018 |
| 7 | Dr V Narendran<br>Dr George Manayath | Aravind Eye Hospital, Coimbatore, Tamil Nadu | Aravind Medical Research Foundation, Institutional Ethics Committee<br>Number: ECR/182/Inst/TN/2013<br>IRB2018010BAS<br>Date of approval: 18 Aug 2018 |
| 8 | Dr Giridhar Anantharaman<br>Dr Mahesh Gopalakrishnan | Giridhar Eye Institute, Cochin, Kerala | Giridhar Eye Institute, Ethics Committee<br>IEC protocol no: 36/2018<br>Date of approval: 13 June 2018 |
| 9 | Dr Sundaram Natarajan<br>Dr Radhika Krishnan | Aditya Jyot Hospital, Mumbai, Maharashtra | Aditya Jyot Eye Hospital, Ethics Committee<br>Date of approval: 30 Aug 2018 |
| 10 | Dr Sheena Liz Mani | Dr Tony Fernandez Eye Hospital, Aluva, Kerala | Dr Tony Fernandez Eye Hospital, Ethics Committee<br>Date of approval: 21 June 2018 |
| 11 | Dr Manisha Agarwal | Dr Shroff's Charity Eye Hospital, New Delhi | Dr Shroff's Charity Eye Hospital, Ethics Committee<br>Date of approval: 29 January 2018 |
| 12 | Dr Tapas Padhi<br>Dr Umesh Behera | LV Prasad Eye Institute, Bhubaneshwar, Odisha | LV Prasad Eye Institute, Ethics Committee<br>Date of approval :10 October 2018 |
| 13 | Dr Harsha Bhattacharjee<br>Dr Manabjyoti Barman | Sri Sankaradeva Nethralaya, Guwahati, Assam | Sri Sankaradeva Nethralaya, Institutional Ethics Committee<br>Reference number: SSN/IEC/OCTOBER/2018/09<br>Date of approval: 8 October 2018 |
| 14 | Dr Gajendra Chawla | Vision Academy–The Socio Medical Society, Bhopal, Madhya Pradesh | Vision Research Foundation, Institution Review Committee<br>Approval number: 674A-2018-P<br>Date of approval: 22 March 2018 |
| 15 | Dr Alok Sen | Sadguru Netra Chikitsalaya, Chitrakoot, Madhya Pradesh | Vision Research Foundation, Institutional Review Committee<br>Approval number: 674A-2018-P<br>Date of approval: 22 March 2018 |
| 16 | Dr Moneesh Saxena | Aurobindo Nethralaya, Raipur, Chhattisgarh | Shri Aurobindo Medical Research Centre, Institutional Review Board<br>Date of approval: 22 June 2018 |

Continued

| Site no. | Name of principal investigator | Hospital name | Ethics approval and date |
|---|---|---|---|
| | **Table 2** Continued | | |
| 17 | Dr Asim K Sil<br>Dr Subhratanu Chakabarty | Netra Niramay Niketan, Haldia, West Bengal | Vivekendra Mission Asram Netra Niramay Niketan, Institutional Review Board<br>Date of approval: 4 September 2018 |
| 18 | Dr Thomas Cherian<br>Dr Reesha KR | Little Flower Hospital and Research Centre, Angamaly, Kerala | Little Flower Hospital and Research Centre, Ethics Committee<br>Date of approval:4 June 2018 |
| 19 | Dr Rushikesh Naigaonkar<br>Dr Abishek Desai | Ganapathy Nethralaya, Jalna, Maharashtra | Shri Ganapati Netralaya, Institutional Ethics Committee<br>Date of approval: 28 July 2018 |
| 20 | Dr Col Madan Deshpande<br>Dr Sucheta Kulkarni | HV Desai Hospital, Pune, Maharashtra | PBMA's H V Desai Eye Hospital, Institutional Review Committee<br>Number: HVD/EC/17/2018<br>Date of approval :21st June 2018 |

screening and associated retinal images, blood and urine tests on the rates of true and false positives and negatives, the characteristics of people with diabetes and its complications, and their quality of life; (2) the data collected through the study on the cost per person of this screening and its cost per person with diabetes, and the costs of clinic visits and treatments for DR; (3) the data and information from past studies on the incidence rates by age and gender of diabetes, DR and other complications of diabetes, transition rates between different stages of the disease and disease-specific mortality rates; and (4) the data from past studies on the costs of care for people with varying severities of DR and other complications of diabetes and on their quality of life. For those variables on which data cannot be collected in this study or obtained from past studies, expert views will be sought, and sensitivity analyses conducted.

The modelling will comprise development of Markov models to track people aged 40 years and above (a) through incidence of diabetes, any DR, STDR, severe visual impairment/blindness and (b) through incidence of diabetes, mild complications other than DR and severe complications other than DR. For each disease state, the models will contain estimates of average annual costs of care and average EQ-5D quality of life. The design of the models will be developed in the light of data availability.

The models will be used to estimate lifetime costs and quality of life (monetised quality adjusted life years, QALYs) from age 40 years and above (a) where the planned screening approach (or approaches) is conducted and necessary treatment given shortly after screening; (b) where the 'gold standard' screening approach is conducted and necessary treatment given shortly after screening; and (c) where no screening is conducted and no treatment given until symptoms develop. The incremental cost effectiveness of the screening in comparison with 'gold standard' screening will be estimated by comparing (a) and (b); and its incremental cost effectiveness in comparison with no screening will be estimated by comparing (a) and (c). A wide range of sensitivity analysis

will be conducted, and a variety of discount rates may be applied.

We will also evaluate and compare the cost effectiveness of retinal photography for everyone with diabetes versus retinal photography only for people with diabetes with suspected high risk of DR, to be developed through the statistical modelling. We will develop a health economics plan after reviewing available data. As an example, Rachapelle et al[31] used a WHO-recommended approach for a cost-effectiveness threshold in their study of the cost utility of telemedicine to screen for DR in India. Under that approach, the interventions costing less than per capita gross domestic product (GDP) per QALY were considered very cost effective, interventions between one time and three times GDP were considered cost effective and interventions more than three times GDP were not considered cost effective.

### Process evaluation
A detailed process evaluation plan will be developed to evaluate the holistic screening for all complications of diabetes, including the teleophthalmology. For each quantitative outcome measure, we will systematically embed qualitative measures in each RE-AIM (reach, efficacy, adoption, implementation and maintenance) dimension to evaluate the implementation strategy of community screening with minimally invasive tests.[32 33]

### Outcomes
The primary outcome is the correlation of RBG levels and POC HbA1c levels. Secondary outcomes include the cut-off value of RBG to define prediabetes; diagnostic accuracy of risk stratification models for diabetes; prevalence and risk stratification for screening for DR; risk model for those at risk of complications of diabetes; identification of cost-effective diagnostic model for diabetes, prediabetes and complications of diabetes and process evaluation of minimally invasive community screening for diabetes and its complications.

## Patient and public involvement

No patient involved.

## Ethics and dissemination

The Indian Council of Medical Research's Health Ministry Screening Committee (HMSC/2018–0494; dated: 17 December 2018) and the institutional ethical committees of all the participating institutions have approved the study (table 2). The main ethical issues in relation to this study are the identifications of people with risk factors for prediabetes, diabetes and its complications. However, the benefits of early diagnosis outweigh these risks. Participants who screen positive for any risk factors will be advised about referral to the local hospitals for treatment. Any breach of confidentiality will be minimised by anonymising participant identifiable information.

The results will be published in open access peer-reviewed journals, presented at scientific meetings and shared with the funder, and specific communication will be organised to target health professionals, policy decision-makers, regulatory bodies and commercial bodies for development of better predictive devices. The anonymised study data will be analysed by the statistical team in the UK. Anonymised patient level data access will be made available to researchers from appropriate data archive for sharing purposes following publication of the study.

**Author affiliations**
[1]NIHR Biomedical Research Centre, Moorfields Eye Hospital NHS Foundation Trust, London, UK
[2]Vision Sciences, UCL Institute of Ophthalmology, London, UK
[3]Retina Department, Vision Research Foundation, Sankara Nethralaya, Chennai, India
[4]Deaprtment of Diabetes, Madras Diabetes Research Foundation, Dr Mohan's Diabetes Specialities Centre, Chennai, India
[5]Retina Department, Hyderabad Eye Research Foundation, LV Prasad Eye Institute, Hyderabad, India
[6]Retina Department, Aravind Medical Research Foundation, Madurai, India
[7]Nightingale-Saunders Clinical Trials and Epidemiology Unit, King's College London, London, UK
[8]Centre for Health Service Economics and Organisation, University of Oxford, Oxford, UK
[9]Institute for Connected Communities, University of East London, London, UK
[10]Department of Ophthalmology, National University Hospital, Singapore
[11]Department of Diabetes, University Hospital, Birmingham, UK

**Acknowledgements** The authors would like to thank all the SMART India collaborators, including fieldworkers, each centre staff, reading centre staff, Jitendra Pal Thethi for the study database and the study participants.

**Contributors** SMART India writing group: SS, RRaman, RRajalakshmi, VM, DM, TD, KR, ATP, RW, GN, GL, WH, DC, JR, JS and RRamakrishnan). Conceptualisation: SS, RRaman, RRajalakshmi, VM, KR, DM, JR, ATP, GL, TD, GN and RW; methodology: SS, TD, GN, RW, ATP, JR, DM and RRamakrishnan; formal analysis: SS, GN, RW, ATP and RRamakrishnan; writing and original draft preparation: SS, DC, RRaman, RRajalakshmi, TD, GN, RW, JS, WH, JR and ATP; writing, and review and editing: SS, RRaman, RRajalakshmi, VM, KR, GL, TD, GN, RW, JR, ATP, WH, DM, DC and RRamakrishnan; funding acquisition: SS, RRaman, RRajalakshmi, VM, KR, TD, GN and RW. Table 2: on behalf of the SMART India Collaborators.

**Funding** This work was supported by Global Challenges Research Fund and UK Research and Innovation through the Medical Research Council (grant number MR/P027881/1). This funding source had no role in the design of this study and will not have any role during its execution, analyses, interpretation of the data or decision to submit results.

**Map disclaimer** The depiction of boundaries on this map does not imply the expression of any opinion whatsoever on the part of BMJ (or any member of its group) concerning the legal status of any country, territory, jurisdiction or area or of its authorities. This map is provided without any warranty of any kind, either express or implied.

**Competing interests** None declared.

**Patient consent for publication** Not required.

**Provenance and peer review** Not commissioned; externally peer reviewed.

**ORCID iDs**
Sobha Sivaprasad http://orcid.org/0000-0001-8952-0659
Rajiv Raman http://orcid.org/0000-0001-5842-0233
A Toby Prevost http://orcid.org/0000-0003-1723-0796

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
