## [Reviewer comments · BMJ Open]

ARTICLE DETAILS

TITLE (PROVISIONAL)	Protocol on a multi-centre statistical and economic modelling study of risk-based stratified and personalised screening for diabetes and its complications in India (SMART India).
AUTHORS	Sivaprasad, Sobha; Raman, Rajiv; Rajalakshmi, Ramachandran; Mohan, V; Deepa, Mohan; Das, Taraprasad; Ramasamy, Kim; Prevost, A Toby; Wittenberg, Raphael; Netuveli, Gopalakrishnan; Lingam, Gopal; Hanif, Wasim; Ramakrishnan, Radha; Ramu, Jayashree; Surya, Janani; Conroy, Dolores

VERSION 1 – REVIEW

REVIEWER	Ali Timucin Atayoglu Medipol University, Turkey
REVIEW RETURNED	31-May-2020

GENERAL COMMENTS	This study has an aim to develop practical and affordable models to diagnose people with diabetes and pre-diabetes and identify those at risk of diabetes complications so that these models can be applied to the population in low and middle-income countries. The use of HbA1c test has several advantages over that of blood glucose, but the higher cost of the test may limit its utility. It may also be impractical in places where reliable test methods of HbA1c are not available. Therefore, the feasibility of using RBG in place of HbA1c may be more advantageous and practical. In order to overcome this limitation, it is important to assess the cut-off values for RBG corresponding to the HbA1c values. However, previous research noted that an RBG cut-off value did not show an acceptable level of sensitivity and specificity, and hence could not be used to define prediabetes. 1. Such previous studies should be mentioned in the introduction with their limitations.2. The limitations and strengths of the current protocol should be discussed in comparison with the previous studies.3. A completed SPIRIT checklist should be included.
---

REVIEWER	Cristina Rolim Neumann Universidade Federal do Rio Grande do Sul Faculdade de medicina
REVIEW RETURNED	15-Jun-2020

GENERAL COMMENTS	This is an ambitious study that aims to develop the best model to assess the presence of diabetes and pre-diabetes using standardized questionnaires and fingertip testing in a population of 20 provinces in India and that could be used in other Low and middle income countries (LMIC). The secondary outcomes include
---

	RBG cut-off for definition of pre-diabetes; diagnostic accuracy of cost-effective risk stratification models for diabetic retinopathy (DR); and models for identifying those at risk of complications of diabetes. For this objective, a large sample from 48000 people will be collected at random. The details of the study as stratify of population, quality control of the tests and data collected, statistical analysis are adequately described. I would have minor doubts: In the description of the study, the authors describe it as a cohort, but with regard to the main objective it seems to me more of a cross-sectional study, and a hypothetical cohort as to the objective of cost effectiveness. Perhaps it would be good to clarify this in the Study Design Section (page 9 line 3-6) because it gives the impression that there will be a follow-up beyond modeling, but reading the rest of the proposal is not mentioned. The author mentions the STARD guideline for the description of studies of diagnostic accuracy, and the description is in accordance with the guideline. I noticed only a small inconsistency in the cutoff point for pre-diabetes in glycated hemoglobin is 6.0 but its reference is ADA- Standards of Medical Care in Diabetes where the cutoff is 5.7. The statistics regarding diagnostic tests are adequately described. Regarding the construction of the Markov model for cost effectiveness, the description of the statistics also seemed correct although personally I have little experience in creating these models. In general, I consider it an admirable proposal that has quality for publication.
--	--

VERSION 1 – AUTHOR RESPONSE

Reviewer 1:

This study has an aim to develop practical and affordable models to diagnose people with diabetes and pre-diabetes and identify those at risk of diabetes complications so that these models can be applied to the population in low and middle-income countries. The use of HbA1c test has several advantages over that of blood glucose, but the higher cost of the test may limit its utility. It may also be impractical in places where reliable test methods of HbA1c are not available. Therefore, the feasibility of using RBG in place of HbA1c may be more advantageous and practical. In order to overcome this limitation, it is important to assess the cut-off values for RBG corresponding to the HbA1c values. However, previous research noted that an RBG cut-off value did not show an acceptable level of sensitivity and specificity, and hence could not be used to define prediabetes.

1. Such previous studies should be mentioned in the introduction with their limitations.

We have added the previous studies and its limitations to the introduction.

2. The limitations and strengths of the current protocol should be discussed in comparison with the previous studies.

We have now added this to the introduction and discussion.

3. A completed SPIRIT checklist should be included.

We have included the SPIRIT checklist.

Reviewer: 2

Reviewer Name

Cristina Rolim Neumann

Institution and Country

Universidade Federal do Rio Grande do Sul

Faculdade de medicina

Please state any competing interests or state 'None declared':

None declared

Please leave your comments for the authors below

This is an ambitious study that aims to develop the best model to assess the presence of diabetes and pre-diabetes using standardized questionnaires and fingertip testing in a population of 20 provinces in India and that could be used in other Low and middle income countries (LMIC). The secondary outcomes include RBG cut-off for definition of pre-diabetes; diagnostic accuracy of cost-effective risk stratification models for diabetic retinopathy (DR); and models for identifying those at risk of complications of diabetes. For this objective, a large sample from 48000 people will be collected at random. The details of the study as stratify of population, quality control of the tests and data collected, statistical analysis are adequately described.

I would have minor doubts: In the description of the study, the authors describe it as a cohort, but with regard to the main objective it seems to me more of a cross-sectional study, and a hypothetical cohort as to the objective of cost effectiveness. Perhaps it would be good to clarify this in the Study Design 1. We have amended this to cross-sectional study.

Section (page 9 line 3-6) because it gives the impression that there will be a follow-up beyond modeling, but reading the rest of the proposal is not mentioned.

2. Our intention is to follow-up the patients referred for treatment of sight threatening diabetic retinopathy for at least their first visit but preferably 3 months. This is now clarified in the text.

The author mentions the STARD guideline for the description of studies of diagnostic accuracy, and the description is in accordance with the guideline. I noticed only a small inconsistency in the cutoff point for pre-diabetes in glycated hemoglobin is 6.0 but its reference is ADA-Standards of Medical Care in Diabetes where the cutoff is 5.7.

This has been corrected to add WHO reference for cut-off 6.0. Thank you.

The statistics regarding diagnostic tests are adequately described. Regarding the construction of the Markov model for cost effectiveness, the description of the statistics also seemed correct although personally I have little experience in creating these models.

In general, I consider it an admirable proposal that has quality for publication.

Thank you.

VERSION 2 – REVIEW

REVIEWER	Ali Timucin Atayoglu Medipol University, Turkey
REVIEW RETURNED	21-Aug-2020
GENERAL COMMENTS	I appreciate the answers of the authors and consider this article has the quality for publication.